# Generation of an Adequate Perfusion Network within Dense Collagen Hydrogels Using Thermoplastic Polymers as Sacrificial Matrix to Promote Cell Viability

**DOI:** 10.3390/bioengineering9070313

**Published:** 2022-07-14

**Authors:** Marie Camman, Pierre Marquaille, Pierre Joanne, Onnik Agbulut, Christophe Hélary

**Affiliations:** 1Laboratoire de Chimie de la Matière Condensée de Paris, UMR 7574, CNRS, Sorbonne Université, F-75005 Paris, France; marie.camman@sorbonne-universite.fr (M.C.); pierre.marquaille@espci.fr (P.M.); 2Biological Adaptation and Ageing, Inserm U1164, UMR 8256, CNRS, Institut de Biologie Paris-Seine (IBPS), Sorbonne Université, F-75005 Paris, France; pierre.joanne@sorbonne-universite.fr

**Keywords:** thermoplastic polymers, HIPS, sacrificial matrix, perfusion, dense collagen hydrogel

## Abstract

Dense collagen hydrogels are promising biomaterials for several tissue-engineering applications. They exhibit high mechanical properties, similar to physiological extracellular matrices, and do not shrink under cellular activity. However, they suffer from several drawbacks, such as weak nutrient and O_2_ diffusion, impacting cell survival. Here, we report a novel strategy to create a perfusion system within dense and thick collagen hydrogels to promote cell viability. The 3D printing of a thermoplastic filament (high-impact polystyrene, HIPS) with a three-wave shape is used to produce an appropriate sacrificial matrix. The HIPS thermoplastic polymer allows for good shape fidelity of the filament and does not collapse under the mechanical load of the collagen solution. After the collagen gels around the filament and dissolves, a channel is generated, allowing for adequate and rapid hydrogel perfusion. The dissolution process does not alter the collagen hydrogel’s physical or chemical properties, and the perfusion is associated with an increased fibroblast survival. Here, we report the novel utilization of thermoplastics to generate a perfusion network within biomimetic collagen hydrogels.

## 1. Introduction

Collagen-based hydrogels have been considered for tissue engineering applications due to their high biocompatibility and bioactivity, as they allow for cell adhesion [1]. However, current fibrillar hydrogels are characterized by poor mechanical and physical properties due to their low initial concentration (<5 mg·mL^−1^) [2]. They contract under cellular activity and become fibrotic-like tissues with a high collagen concentration. Moreover, in cell culture, this shrinking leads to a reduced diffusion of O_2_ and nutrients for the cells, resulting in cell death [3]. When increasing the concentration, collagen hydrogels exhibit high mechanical properties and are stable over several weeks when implanted in vivo [4,5]. In addition, they mimic the physical properties of native extracellular matrices (ECMs), such as dermal and cardiac properties. Two strategies have primarily been used to produce dense collagen hydrogels: (1) the compression of low-concentration gels [6,7] or (2) evaporation [4]. Compression creates high collagen concentrations (>100 mg·mL^−1^), but it can be more difficult to obtain precise concentrations using this method compared with evaporation (<80 mg·mL^−1^). Just as with contracted hydrogels, high collagen concentrations dramatically affect cell survival inside the hydrogel. In both cases, nutrient and O_2_ availability depend on the collagen density and gel thickness [8,9]. For example, fibroblasts die from apoptosis after a 500 µm migration within dense collagen hydrogels [10]. Hence, strategies to combine a high collagen concentration with cell survival must be discovered. In vivo, extracellular matrices are composed of concentrated collagen (around 30 mg·mL^−1^ in the heart [11]), but the cells survive due to a very developed vascularization network. For instance, about 2500 capillaries per mm^2^ can be found in the heart to ensure cell survival [12]. In vitro, this vascularization network can be mimicked by a perfusion system that diffuses the culture medium inside the gel. Using molding needles [13,14] or nylon wires [15,16] within a collagen hydrogel, straight channels with diameters ranging from 75 to 520 µm can be obtained [17]. This technique creates channels that cross the gel from one side to the other for active (when the medium is perfused with a pressure controller) or passive perfusion (when the medium diffuses through the perfusion channel). Usually, these channels diffuse the culture medium without any external pumping setup. The geometry of the network is limited, and the space between the two channels must be below the diffusion limit to ensure cell survival. Nazhat et al. developed a dense collagen hydrogel with 30 µm-long phosphate-based glass fibers randomly dispersed inside the construct [6]. Dense collagen gels were obtained using plastic compression. This process generated a high number of channels at a reduced volume, but which could not be perfused due to their small diameters. However, the development of 3D printing has permitted the production of sacrificial matrices. By printing two different inks within the same construct, one definitive ink dedicated to the walls and one sacrificial ink devoted to the channels, Lee et al. created on-demand channels for perfusion [18]. They found that the sacrificial material had to meet several criteria, such as (1) good mechanical properties to ensure the cohesiveness of the construct, (2) an efficient method of elimination, and (3) a proper elimination procedure that would not affect the global physicochemical and mechanical properties of the construct. For instance, collagen-based hydrogels are sensitive to temperatures above 50 °C; hence, the sacrificial matrix must be removed without heating. The soft materials, gelatin [18] and pluronic [19], were removed at 37 and 4 °C, respectively, without any denaturation of the collagen fibrils. However, these materials lack mechanical strength; hence, the structure may collapse under the collagen weight. Therefore, sacrificial channels made of soft polymers are crushed by the weight of the upper collagen layers, and the final shape differs from the designed one. New sacrificial materials with load-bearing properties must be discovered to create on-demand channels with high printing fidelity.

Here, we report a novel method to construct a perfusion system within highly dense collagen hydrogels (30 mg·mL^−1^). For this purpose, a sacrificial matrix consisting of thermoplastic polymers was produced by 3D printing and set within a dense collagen hydrogel. Fibroblasts were cultivated within hydrogels to assess the impact of the perfusion system on cell viability.

## 2. Materials and Methods

### 2.1. Collagen Extraction and Purification

Type I collagen was extracted and purified from rat tail tendons as previously described [20]. The rat tails were briefly rinsed with 70% ethanol and cut into small 1 cm-long pieces to extract the tendons. The tendons were solubilized in 500 mM acetic acid. After precipitation with 0.7 M NaCl, centrifugation, and dialysis, collagen purity was observed with SDS-PAGE electrophoresis, and its concentration was estimated with hydroxyproline titration. After evaporation in a safety cabinet for several days at room temperature, a collagen solution with a concentration of 30 mg·mL^−1^ was obtained. The collagen concentration was assessed every day using the following equation:$${\left[\mathrm{Collagen}\right]}_{t}=\frac{{\mathrm{weight}}_{t}}{{\mathrm{weight}}_{0}}\times {\left[\mathrm{Collagen}\right]}_{0}\;$$
once the ${\left[\mathrm{Collagen}\right]}_{t}=30$ mg·mL^−1^ collagen solution was stored at 4 °C before utilization.

### 2.2. Design of the Sacrificial Matrix

The different shapes of the sacrificial matrix were designed using AutoDesk Fusion 360 software (San Rafael, CA, USA) to obtain a 3D model (Figure 1). The 3D object file (.stl) was then sliced using Z-suite software (Zortrax SA, Olsztyn, Poland) with a layer thickness of 0.09 mm to obtain high-fidelity printing. First, the grid model was printed with different thermoplastic polymers. Z-PLA, Z-ABS, and Z-HIPS were tested. The test conditions are presented in the following Table 1. These materials were chosen because they withstand the ammonia vapors required for collagen fibrillogenesis and can be eliminated without heating.

### 2.3. Preparation of the Constructs

A plastic mold was designed and printed to place the sacrificial channel in a reproducible manner. Once the sacrificial matrix was set, the concentrated collagen (30 mg·mL^−1^) solution was poured to fill the plastic mold (Figure 2). The collagen gel dimensions were 12 × 12 × 5 mm in the inside compartment. A control construct was obtained by setting three needles (with a 23G-external diameter of 600 µm) into three specific holes to create straight channels, and then the collagen solution was added. The constructs were gelled under ammonia vapors. After several washes in PBS 1X to reach a neutral pH, the plastic molds were removed to place the gel in the appropriate solvent to eliminate the sacrificial ink. The gels were then extensively washed in PBS 1X to remove the solvent. For the control samples, the 3 needles were removed, and the gels were rinsed with PBS prior to utilization.

Three needle-molded (23 G) channels were added at 500 µm above the perfusion channel for cellularized constructs. These channels were then filled with cells.

### 2.4. Differential Scanning Calorimetry

Ten to twenty milligrams of collagen hydrogel was placed into an aluminum pan. Measurements were acquired with NanoScan differential scanning calorimetry. A temperature scan from 20 to 100 °C with a 10 °C·min^−1^ ramp was performed to detect the collagen fibril denaturation peak.

### 2.5. Rheological Measurements

Shear oscillatory measurements were performed on collagen disks cast in 24-well plates with an Anton Paar rheometer. An 8 mm plan geometry was fitted with a rough surface to avoid gel slipping. All measurements were performed at 37 °C. Storage modulus G′ and loss modulus G″ were recorded during a frequency sweep from 0.1 to 10 Hz with an imposed strain of 1%. This strain corresponded to non-destructive conditions as previously checked (data not shown). The gap between the gel and the geometry was set to have a minimal normal force of 0.01 N. Three samples of each matrix were tested.

### 2.6. Scanning Electron Microscopy

The collagen constructs were cross-linked overnight at 4 °C using a 4% paraformaldehyde (PFA) solution (*w*/*v*) in PBS. This step was followed by a 1 h fixation at 4 °C in a 2.5% glutaraldehyde solution diluted in cacodylate buffer. The samples were then dehydrated using ethanol baths with increasing concentrations and then supercritically dried. The samples were coated with a 10 nm gold layer before their observation under a Hitachi S-3400N Scanning Electron Microscope (operating at 5 kV).

### 2.7. MicroCT Imaging

The gels were loaded with a Micropaque contrast agent (Guerbet) before they were observed and scanned using a high-resolution X-ray micro-CT system (Quantum FX Caliper, Life Sciences, Perkin Elmer, Waltham, MA, USA) hosted by a PIV platform (UR2496, Montrouge, France). Standard acquisition settings were applied (voltage: 90 kV, intensity: 160 mA), and the scans were performed with a field of view of 1 cm^2^. The micro-CT datasets were analyzed using a built-in multiplanar reconstruction tool, Osirix Lite (Pixmeo, Geneva, Switzerland), to obtain time-series images and 3D reconstruction.

### 2.8. Diffusion Study

The gels were perfused with blue-colored DMEM (Gibco) at 0.05 mL·min^−1^. The gels were fixed inside a petri dish using agarose 5% wt to maintain the gel and the perfusion needles (Appendix A). The perfusion was ensured with a syringe controller. A 20 mL syringe was loaded with brilliant-blue-colored DMEM (E133). Images were taken every 20 min.

### 2.9. Cell Cultivation

Normal human dermal fibroblasts (NHDFs) were cultured in a complete cell culture medium. Dulbecco’s modified eagle medium (DMEM) was supplemented with a 10% fetal bovine serum, 100 U·mL^−1^ penicillin, 100 μg·mL^−1^ streptomycin, 0.25 μg·mL^−1^ Fungizone, and GlutaMAX. Tissue culture flasks (75 cm^2^) were kept at 37 °C in a 5% CO_2_ atmosphere. Before confluence, the cells were removed from the culture flasks with 0.1% trypsin and 0.02% EDTA treatment. The cells were rinsed and suspended in a complete culture medium before use. Then, after centrifugation, the cell pellet was mixed with pure Matrigel^®^ (Sigma, Bioreagent, Saint-Louis, MO, USA) at 1.10^6^ cells·mL^−1^, and 3 µL was seeded into each large channel made by needles (500 µm). The fibroblasts were cultivated inside the channels for over a month.

### 2.10. Live/Dead Assay

A live/dead assay (Thermofisher, Waltham, MA, USA) was performed to assess cell survival. Two reagents, calcein-AM and ethidium bromide, were added to the culture medium of cellularized dense hydrogels. Cell viability was observed by producing a green, fluorescent molecule created by the metabolization of calcein-AM in the living cells. Ethidium bromide stained the nuclear DNA of dead cells. The gels were incubated in these reagents for 30 min at 37 °C. Subsequently, they were embedded in agarose and cut into 200 µm slices to be observed with fluorescence microscopy.

### 2.11. Statistical Analysis

All experiments were conducted at least twice, and the data are expressed as mean values ± standard deviation (SD). The differences were analyzed using Mann–Whitney tests (when n > 4); *p* < 0.05 was considered significant.

## 3. Results and Discussion

### 3.1. Thermoplastics Screening

Different thermoplastics were tested to obtain well-defined channels with easy removal. The main advantage of thermoplastic materials is their high mechanical property; they retain their shape even when surrounded by dense collagens. They can be eliminated by solvents, but they may affect the collagen fibril structure and hydrogel properties. Thus, we first selected the appropriate polymer and optimized the sacrificial perfusion network inside a dense collagen hydrogel. The first polymer tested was ABS (acrylonitrile butadiene styrene), widely used to make light and robust objects. It can be dissolved by acetone. PLA (polylactic acid) was used due to its ability to be easily shaped with 3D printing and its fast dissolution using dichloromethane. Lastly, HIPS (high-impact polystyrene) was used due to its high solubility in dichloromethane. After a 48-h incubation in their dedicated solvents under stirring to remove sacrificial matrices, a micro-computed tomography analysis was performed (Figure 3, right panels). The needle molding, used as a control, revealed open and well-defined channels without post-treatment (Figure 3A). The ABS matrix solved in acetone was partially removed (after 48 h), as some polymer fragments remained in the channel. In addition, the hydrogel shrank and whitened (Figure 1B), suggesting that there was an alteration of the collagen hydrogel properties.

No shrinkage or whitening was observed with the dichloromethane baths used to dissolve the PLA (Figure 3C). However, PLA is known to degrade into lactic acid [21]. Due to its acidic pH, this product can degrade the collagen gel. It was replaced by a HIPS sacrificial matrix to avoid such a phenomenon. After a 48-h dichloromethane bath, some residues were observed inside the channels with a microCT scan (Figure 3D). Because these residues were liquid, an additional injection of dichloromethane inside the channels was implemented to empty the channels (Figure 3E). Hence, the HIPS matrix was the best candidate to create open and well-defined channels by sacrificial matrix 3D printing. Therefore, this system was used in further experiments.

### 3.2. Dichloromethane Treatment and Collagen Physico-Chemical Properties

Additional experiments were conducted to determine the impact of dichloromethane on the collagen hydrogel structure and its physical properties. For this purpose, different techniques were used to assess the collagen fibril integrity. First, rheology measurements were performed and revealed a 1 kPa decrease in the storage modulus between non-treated gels (G′ = 4, 6 ± 0, 4 kPa) and those treated with dichloromethane (G′ = 3, 6 ± 0, 6 kPa) (Figure 4A). The dichloromethane bath did not significantly affect the loss modulus (non-treated: G″ = 558 ± 77 Pa, treated: G″ = 504 ± 88 Pa) (Figure 4B). Differential scanning calorimetry did not reveal any difference in the temperature denaturation of the fibrils after dichloromethane treatment (Figure 4C,D). Lastly, scanning electron microscopy (SEM) observation revealed similar fibrillar networks with or without dichloromethane treatment (Figure 4E). These methods confirmed that the dichloromethane treatment significantly altered neither collagen fibrils nor hydrogel mechanical properties.

### 3.3. Sacrificial Matrix Design Optimization

Unlike needles, which can only generate straight channels, sacrificial matrices can be produced with different shapes to optimize O_2_ and nutrient diffusion under specific conditions. In this study, the aim was to bring nutrients and O_2_ to the center of the gel to promote cell survival. To do so, matrices were designed with a one- or three-wave shape (Figure 5A) to increase the area covered by the culture medium compared with that of the needles. Perfusion efficacy was related to the hydrogel area exposed to medium diffusion. The 3D matrix models were drawn with a computer with wider extremities to facilitate future perfusion. They were printed with high fidelity and retained their shape throughout the process. After collagen molding and sacrificial matrix removal, micro-computed tomography showed the different geometries obtained (Figure 5B). All the channels were entirely emptied and filled with the contrast agent. The printed matrices covered a larger area on the collagen gel than the needles and were perfused with a single input. Once perfused with a blue-colored solution at 0.5 mL·min^−1^, as described in the literature [14], the solution diffusion inside the gel was observed for over 120 min. The needles revealed a lower diffusion efficiency compared with those of the printed matrices. After two hours of perfusion using the needles, a small fraction of hydrogel was colored (Figure 5, right panel). An image analysis revealed a diffusion speed of around 4 µm·min^−1^ (data not shown). Twenty needle channels should have been set to perfuse the whole hydrogel within 60 min, but this was infeasible. The results were different when the sacrificial matrices were used. After two hours, the entire gel was colored when a single- or three-wave geometry was used (Figure 5). The most promising results were obtained with a three-wave geometry as the diffusion area rapidly increased to perfuse the whole volume within 20 min (Figure 5).

### 3.4. Cell Colonization

First, the cytotoxicity of the channel generation process was assessed by cultivating normal human dermal fibroblasts on top of collagen gels, whether treated with dichloromethane or not. After 48 h, no significant difference in cell survival was observed (Appendix A). Multiple rinsing baths after the dichloromethane incubation efficiently removed any trace of the solvents. The next step was to colonize the hydrogel with cells. Due to the use of chemical solvents, acid collagen, and ammonia vapors, the cells were added after the completed process. Hence, it was required to design a specific strategy to seed the cells. In this study, needle molding created three large channels filled with cells to form cylindrical microtissues (Figure 2 with the optional step). Human dermal fibroblasts, used as a cell model, were then seeded inside these channels with Matrigel^®^ to ensure a 3D colonization (Figure 6, pink channels). Cell seeding within preformed hydrogels is currently used in tissue engineering, and this system could be used to cultivate cardiac or muscle cells to form fibers in a biomimetic extracellular matrix. After 7 days in culture, live/dead staining was performed, and the gel slices were observed (Figure 6). This experiment was conducted on hydrogels with or without perfusion channels (Figure 6, red channels). The fibroblasts underwent proliferation until they reached confluency inside the channel. The number of live fibroblasts (in green) sharply increased when the porosity dedicated to perfusion was present (Figure 6). Cell viability was similar irrespective of the location of the three cellularized channels (Appendix A). These results show that the perfusion channel improved the nutrient diffusion inside the collagen hydrogel, thereby reducing cell death. We found in our study that with such a channel, dense collagen hydrogels (30 mg·mL^−1^) could be colonized in the whole volume, and they did not lead to cell death, as was previously reported in the literature [10].

## 4. Conclusions

We have developed a novel method to create an on-demand perfusion channel in dense collagen hydrogels. Our process involved the design of a stiff 3D-printed sacrificial matrix, and its embedment inside a dense collagen hydrogel possessing high mechanical and physical properties. Due to the porosity, the whole hydrogel volume was perfused within 20 min, increasing cell survival. Our findings offer a simple and versatile method to generate a perfusion network inside collagen and can be used for several tissue-engineering applications, such as vascularized dermis or cardiac tissue modeling.

## Figures and Tables

**Figure 1 bioengineering-09-00313-f001:**
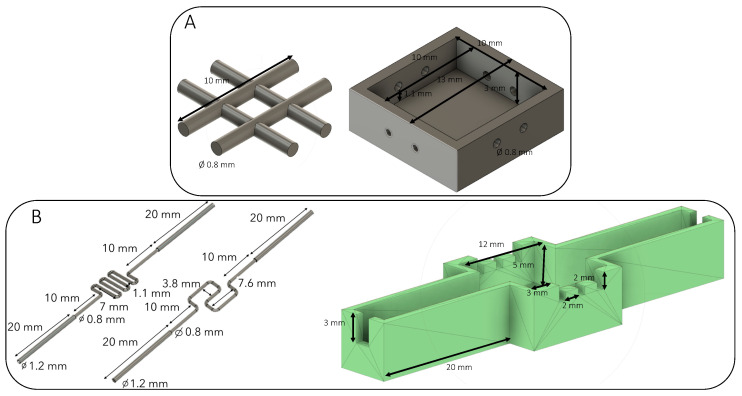
Design of sacrificial shapes and molds used. (**A**) A basic 0.8 mm grid placed in the dedicated mold. (**B**) Two different perfusion channels with their molds. The extremities of the channels were placed in the large notches, whereas the needles were placed inside the three small incisions. The needles were localized 500 µm above the perfusion channel.

**Figure 2 bioengineering-09-00313-f002:**
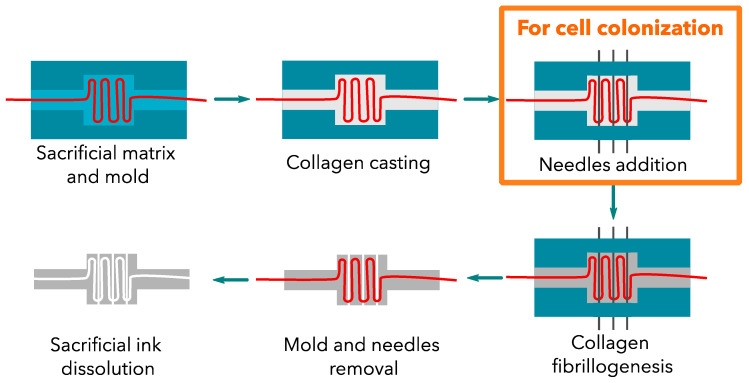
Collagen construct development. The addition of needles is only required for cellularized constructs. These large channels were filled with cells.

**Figure 3 bioengineering-09-00313-f003:**
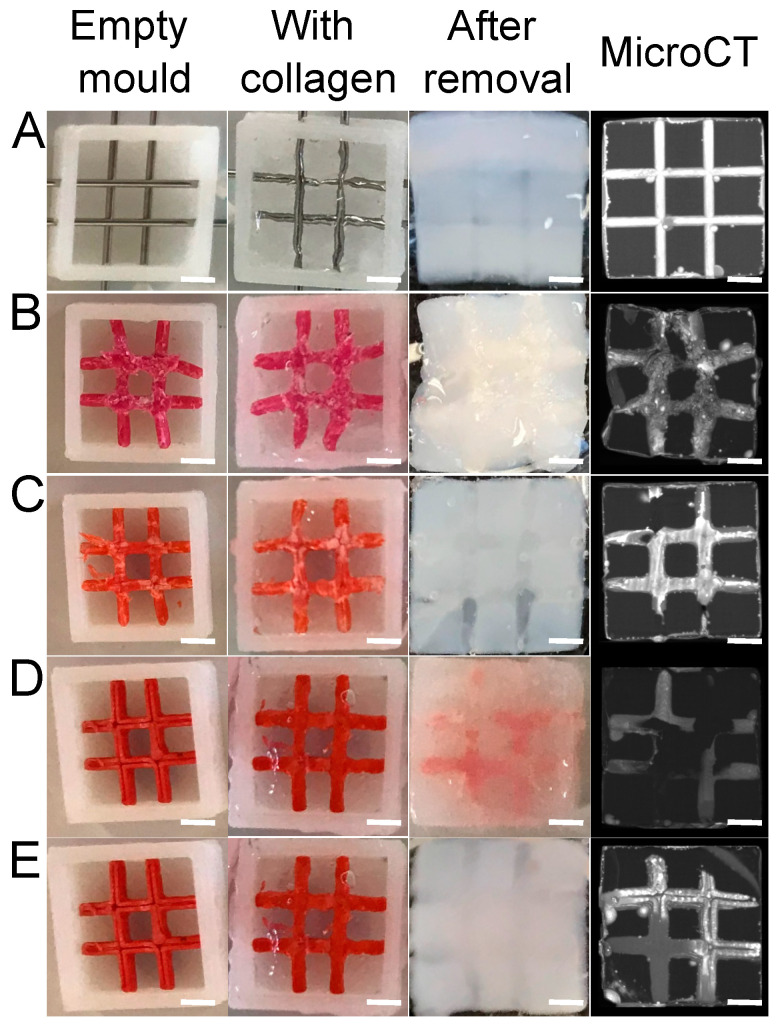
Screening of the different thermoplastics used as a sacrificial matrix. (**A**) Needles, (**B**) ABS, (**C**) PLA, HIPS without (**D**) and with (**E**) dichloromethane injection. Scale bar: 2 mm.

**Figure 4 bioengineering-09-00313-f004:**
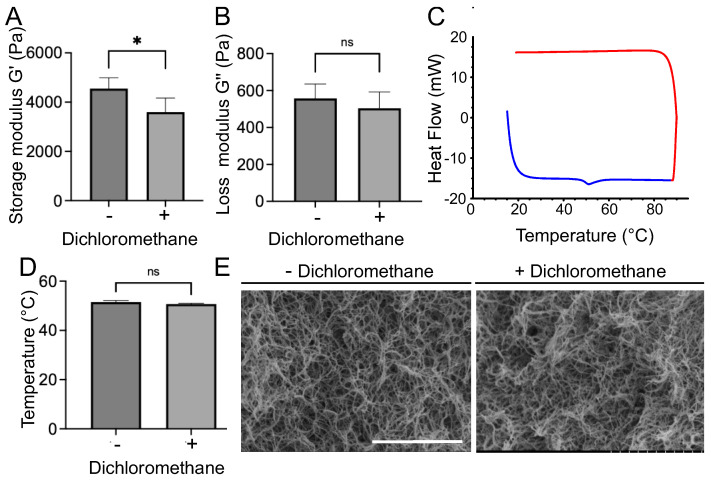
Mechanical and structural characterization of collagen gels before and after dichloromethane treatment. (**A**) Storage modulus and (**B**) loss modulus measured by rheology. (C) Denaturation curve for the sample treated with dichloromethane. The blue curve represents the thermal exchange during the temperature rise, whereas the red curve corresponds to the decrease in temperature. Only one denaturation peak is visible at 55 °C. (**D**) Temperature of denaturation peak of collagen fibrils measured by differential scanning calorimetry (DSC). (**E**) SEM images of collagen fibrils without (−) and with (+) a dichloromethane bath. Scale bar: 5 µm. * corresponds to *p* < 0.05, ns *p* > 0.05.

**Figure 5 bioengineering-09-00313-f005:**
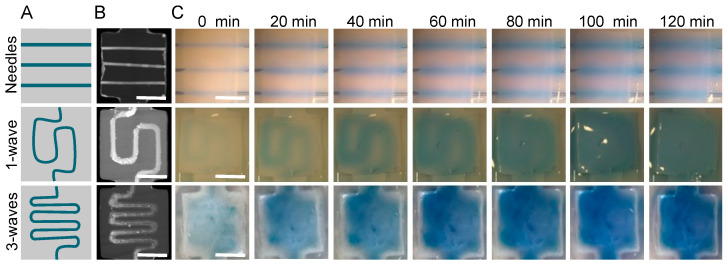
Medium diffusion perfusion channels made with three different geometries. (**A**) Perfusion channel design. (**B**) MicroCT imaging. (**C**) Medium diffusion over time when perfused at 0.05 mL·min^−1^. First line: 3 needles (23 G 600 µm external diameter). Second line: 1 wave geometry made with a sacrificial matrix of HIPS. Third line: three-waves geometry made with a sacrificial matrix of HIPS. All gels were perfused at 0.05 mL·min^−1^ and followed every 20 min for 2 h. Scale bar: 5 mm.

**Figure 6 bioengineering-09-00313-f006:**
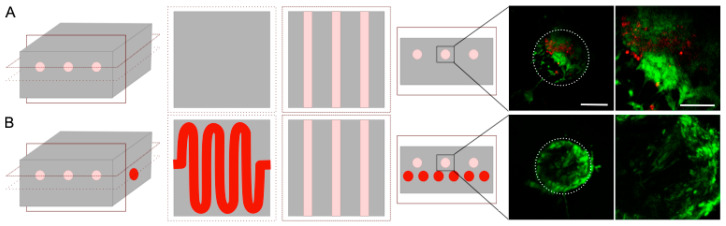
Live (green)/dead (red) staining after 7 days of cultivation inside collagen hydrogel without (**A**) or with (**B**) a passive perfusion channel. Slices were obtained in the middle of each construct. The white circle represents the channel walls from a side view. Scale bar: 250 and 50 µm.

**Table 1 bioengineering-09-00313-t001:** Sacrificial material tested and their associated solvents.

Sacrificial Material	Full Name	Dissolution
Z-PLA	PolyLacticAcid	NH3
		Dichloromethane
Z-ABS	Acrylonitrile butadiene styrene	Acetone
		Dichloromethane
Z-HIPS	High impact polystyrene	Dichloromethane

## Data Availability

Not applicable.

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
