# Peer review of "Generation of an Adequate Perfusion Network within Dense Collagen Hydrogels Using Thermoplastic Polymers as Sacrificial Matrix to Promote Cell Viability"

_bioengineering, 2022, doi:10.3390/bioengineering9070313_

Round 1

Reviewer 1 Report

All comments and issues from the previously suggested review were addressed in full.

Author Response

Thank you.

Reviewer 2 Report

The authors have made great efforts to improve this paper. It can be published on bioengineering now.

Author Response

We thank the reviewer to have accepted our manuscript for publication in Bioengineering.

Reviewer 3 Report

The communication article by Camman et al. describes the feasibility of using thermoplastic polymers as sacrificial entities when generating perfusable channels inside dense collagen matrices. Even though this technique holds promise the authors need to provide additional clarification to be considered for publication.

1.     The authors mention in the abstract that collagen does not shrink under cellular activity. This statement is not entirely correct as depending on the collagen concentration and the cell type, collagen is known to contract. For example a certain density of fibroblasts when cultured in collagen causes matrix compaction and shrinkage.

2.     Why is it important to make such dense collagen hydrogels? How is it better than other techniques or using low dense gels for biological applications? The authors are requested to provide the rationale behind using such dense hydrogels. What do they mean by dense? Some quantifiable numbers should be mentioned.

3.     Why did the authors choose thermoplastic polymers? Some rationale and background for choosing this over other materials need to be provided. The authors mention that pluronic or gelatin will collapse under the weight of the collagen gel but there are studies using nanoclay reinforced pluronic where perfusable channels were fabricated inside PDMS [Kui Zhou et al 2021 Biomed. Mater. 16 045005]. Have the authors tried such techniques or can they comment if such materials could also be used with their collagen?

4.     In the materials methods section, the authors mention they perform an evaporation step under the biosafety hood.  Can they also provide specific details for the readers regarding this step? How many days and what conditions did they follow for the evaporation? How was the evaporation monitored? How was the concentration of collagen measured after evaporation?

5.     In line 240: the authors mention that ‘Needles revealed a lower diffusion efficiency 240 compared to printed matrices’. Why did this happen? If the needles were removed after channel fabrication and perfused as they did for the other channels, it can be assumed that the solution being perfused will diffuse throughout the collagen matrix after some time.

6.     In section 3.4. Can the authors provide some quantification for the cell viability? Some Cell viability and proliferation assays such as CCK8 or Alamar Blue would be helpful to substantiate their claims. Also, the authors are requested to provide some higher magnification images for the cells as it is not clear what the images are trying to indicate. Further, these images should be incorporated into the main manuscript as they imply the application of their techniques in studying biological phenomenon.  

Author Response

Comment 1: The authors mention in the abstract that collagen does not shrink under cellular activity. This statement is not entirely correct as depending on the collagen concentration and the cell type, collagen is known to contract. For example, a certain density of fibroblasts when cultured in collagen causes matrix compaction and shrinkage.

Answer 1:  We apologize if we have not been very clear in our explanations in the manuscript but we would like to say “dense” collagen hydrogels do not shrink under cellular activity. We mean hydrogels fabricated with a collagen concentration over 30 mg.mL-1. This phenomenon is clearly described in our following publications:

  • doi: 10.1016/j.biomaterials.2004.05.016
  • doi: 10.1016/j.biomaterials.2006.04.005.

Comment 2: Why is it important to make such dense collagen hydrogels? How is it better than other techniques or using low dense gels for biological applications? The authors are requested to provide the rationale behind using such dense hydrogels. What do they mean by dense? Some quantifiable numbers should be mentioned.

Answer 2:  As mentioned in the introduction (see below), dense are characterized by a collagen concentration over 30 mg.mL-1. The advantage of this dense hydrogels is their stability and a tunable concentration (from 30 to 60 mg.mL-1). In addition, they mimic the native extracellular matrices. As an example, a 30 mg.mL-1 collagen hydrogels mimic the physical properties of the cardiac extracellular matrix (Young modulus around 10 kPa). This information is now included in the introduction: “By increasing the concentration, collagen hydrogels exhibit high mechanical properties and are stable over several weeks when implanted in vivo [4,5]. In addition, they mimic the physical properties of native extracellular matrices (ECM) such as dermis or cardiac ones”.

Moreover, the advantage over the low concentrated collagen hydrogels is dense collagen hydrogels do not contract. This information is now included in the introduction: “Unfortunately, current fibrillar hydrogels are characterized by poor mechanical and physical properties due to their low initial concentration (< 5 mg.mL-1) [2]. They contract under the cellular activities and become fibrotic-like tissues with a high collagen concentration.”

Comment 3: Why did the authors choose thermoplastic polymers? Some rationale and background for choosing this over other materials need to be provided. The authors mention that pluronic or gelatin will collapse under the weight of the collagen gel but there are studies using nanoclay reinforced pluronic where perfusable channels were fabricated inside PDMS [Kui Zhou et al 2021 Biomed. Mater. 16 045005]. Have the authors tried such techniques or can they comment if such materials could also be used with their collagen?

Answer 3:  As mentioned in the “results and discussion” section, thermoplastics were used because they exhibit high mechanical properties and allow for a very good shape fidelity of the generated channels. Compared to pluronic and gelatin, the channels don’t collapse. Unlike sacrificial matrices made with nanoclay reinforced pluronic, the fabrication is simple and cost effective. It can be done with a regular 3D printer.

This information is now included in the manuscript: “Different thermoplastics were tested to obtain well-defined channels with easy removal. The main advantages of thermoplastic materials are their high mechanical properties: they keep their shape even once surrounded by dense collagen. They can be eliminated by solvents, but they must not affect the collagen fibril structure and hydrogel properties. Thus, we first selected the appropriate polymer and optimized the sacrificial perfusion network inside a dense collagen hydrogel.”

Comment 4: In the materials methods section, the authors mention they perform an evaporation step under the biosafety hood. Can they also provide specific details for the readers regarding this step? How many days and what conditions did they follow for the evaporation? How was the evaporation monitored? How was the concentration of collagen measured after evaporation?

Answer 4:  Actually, the evaporation time depends on the volume of dense collagen solution you want to get at the end. So, we have not a general answer. Regarding the calculation, we enclose the formula:

Comment 5: In line 240: the authors mention that ‘Needles revealed a lower diffusion efficiency 240 compared to printed matrices’. Why did this happen? If the needles were removed after channel fabrication and perfused as they did for the other channels, it can be assumed that the solution being perfused will diffuse throughout the collagen matrix after some time.

Answer 5:  In figure 5, we clearly see the liquid has not diffused through the entire hydrogel volume. Diffusion is intrinsically linked with the area of contact between the medium and the gel. In the case of the needles, the area is reduced compared to the use of sacrificial matrices. Hence, this evidences that the liquid diffusion from the channels made with needles is slower.

Moreover, the needles have been removed before perfusion. It is the perfusion of the three channels that is shown in Figure 5. A sentence has been added in the section 2.3 “For the control samples, the 3 needles were removed and the gels rinsed with PBS.”

Comment 6: In section 3.4. Can the authors provide some quantification for the cell viability? Some Cell viability and proliferation assays such as CCK8 or Alamar Blue would be helpful to substantiate their claims. Also, the authors are requested to provide some higher magnification images for the cells as it is not clear what the images are trying to indicate. Further, these images should be incorporated into the main manuscript as they imply the application of their techniques in studying biological phenomenon.

Answer 6:  The live dead assay is the adequate test to monitor the fraction of dead cells. The metabolic assays such as CCK8 or Alamar blue have some limitations because they measure the cellular metabolic activity. For instance, a metabolic activity could be increased because the cells are stressed, but it does not mean the number of cells has increased.

Regarding the images, we have now encompassed higher magnification images.

Reviewer 4 Report

This manuscript presented by Camman et al. reports the fabrication of perfusion network in collagen hydrogels by using thermoplastic polymers as sacrificial matrix to promote cell viability. The authors utilized 3D printing to fabricate a 3 waved thermoplastic polymer scaffold as the sacrificial matrix; then, collagen solutions were casted to generate the perfusion network. The polymer matrix was removed by dichloromethane, and the fibroblasts were cultured inside the collagen gels through the needle channels for viability assessment. The authors demonstrated the advantage of using thermoplastic polymer as the sacrificial matrix to create dense collagen perfusion network. The contents on collagen network for 3D cell culture is of general interest for tissue engineering community and suitable for Bioengineering scope. The experiments are well designed, and the conclusions are well supported by the data. I suggest minor revisions for publication. Here are my questions to the authors:

1.       1. Line 58, why plastic compression increases the number of channels?

2.       2. Do the fibroblasts undergo proliferation within the network? If so, when will they reach the stationary phase?

3.       3. I am curious about if there are any micro/nano-structures on the collagen hydrogel surface, since there are reports showing nano-topography can dictate the proliferation and phenotyping of fibroblasts (https://www.sciencedirect.com/science/article/abs/pii/S1549963421000083).  

4.       4. Is it possible to start with even higher collagen concentration (e.g. > 50 mg/mL) for creating the perfusion network?

5.       5. The storage modulus is decreased with DCM treatment (Fig 4A); what does this property imply and how does it impact the network stability and cell viability?

6.       6. Fig 4C lacks of description, what are the blue and red curves indicated?

7.       7. In Fig 4E, the right image seems like the zoomed in left image. In that case, the left one can be removed to minimize redundancy.

Author Response

Comment 1: Line 58, why plastic compression increases the number of channels?

Answer 1:  In this article, the authors begin with a low concentrated collagen hydrogel and a limited number of glass fibers. After plastic compression, the final volume of the collagen hydrogel is reduced but the number of glass fibers is identical resulting in a higher density of pores compared to others techniques. The manuscript has been changed to be clearer.

Comment 2: Do the fibroblasts undergo proliferation within the network? If so, when will they reach the stationary phase?

Answer 2:  Despite the fibroblasts are seeded with a large density within the channels, they proliferate and fill the channel to reach a kind of confluency in 3D.

Comment 3: I am curious about if there are any micro/nano-structures on the collagen hydrogel surface, since there are reports showing nano-topography can dictate the proliferation and phenotyping of fibroblasts (https://www.sciencedirect.com/science/article/abs/pii/S1549963421000083).

Answer 3:  With a 30 mg.mL-1 collagen concentration, hydrogels do not exhibit a specific nano-topography. Moreover, at the beginning, cells are embedded in a loose matrix made of Matrigel®. So, the impact of the mechanical properties of the dense matrix would be only seen after the Matrigel remodeling by cells.

Comment 4: Is it possible to start with even higher collagen concentration (e.g. > 50 mg/mL) for creating the perfusion network?

Answer 4:  With the technique of evaporation the maximal concentration which can be reached is 60 mg.mL-1. Above collagen solution are too viscous to be used. We chose 30 mg.mL-1 to mimic the native cardiac extracellular matrix.

Comment 5: The storage modulus is decreased with DCM treatment (Fig 4A); what does this property imply and how does it impact the network stability and cell viability?

Answer 5:  It is true the storage modulus slightly decreased with the dichloromethane treatment. This small difference could be explained by the destruction of certain hydrophobic bonds between fibrils. However, the network stability is slightly impacted as the decrease is less than 20%. In addition, the storage modulus is still close to that of native extracellular matrices.

Comment 6: Fig 4C lacks of description, what are the blue and red curves indicated?

Answer 6:  The legend has been changed accordingly. The blue curve corresponds to the thermal exchanges during the rise of temperature and the red one to the decrease in temperature.

Comment 7: In Fig 4E, the right image seems like the zoomed in left image. In that case, the left one can be removed to minimize redundancy.

Answer 7:  We think it is important to keep the zoomed images to have a more precise view of the collagen ultrastructure. We removed the large frame images to avoid redundancy.

Round 2

Reviewer 3 Report

The authors have addressed all my comments.